# On the Effect of Channel Knowledge in Underwater Acoustic Communications: Estimation, Prediction and Protocol

Andrea Petroni [1], Gaetano Scarano [2], Roberto Cusani [2] and Mauro Biagi [2,*]

1   Fondazione Ugo Bordoni, 00184 Rome, Italy
2   Deptartment of Information, Electronics and Telecommunications (DIET) Engineering,
    Sapienza University of Rome, Via Eudossiana 18, 00184 Rome, Italy
*   Correspondence: mauro.biagi@uniroma1.it

**Abstract:** Underwater acoustic communications are limited by the following channel impairments: time variability, narrow bandwidth, multipath, frequency selective fading and the Doppler effect. Orthogonal Frequency Division Modulation (OFDM) is recognized as an effective solution to such impairments, especially when optimally designed according to the propagation conditions. On the other hand, OFDM implementation requires accurate channel knowledge atboth transmitter and receiver sides. Long propagation delay may lead to outdated channel information. In this work, we present an adaptive OFDM scheme where channel state information is predicted through a Kalman-like filter so as to optimize communication parameters, including the cyclic prefix length. This mechanism aims to mitigate the variability of channel delay spread. This is cast in a protocol where channel estimation/prediction are jointly considered, so as to allow efficiency. The performance obtained through extensive simulations using real channels and interference show the effectiveness of the proposed scheme, both in terms of rate and reliability, at the expense of an increasing complexity. However, this solution is significantly preferable to the conventional mechanism, where channel estimation is performed only at the receiver, with channel coefficients sent back to the transmit node by means of frequent overhead signaling.

**Keywords:** underwater acoustic communications; OFDM; interference; estimation; prediction; Kalman filtering

## 1. Introduction

In spite of their long and consolidated history, underwater acoustic communications (UWACs) have recently experienced a peak of interest from the scientific community, as they represent an effective technology for a wide range of applications [1]. Real-time control and communication with underwater remote instrumentation and autonomous vehicles [2], coastal surveillance [3] and shipping traffic management [4], environmental monitoring [5] and data collection [6] are some of the most widespread activities where the use of UWACs has been shown to be suitable.

Regarding communication aspects, UWAC links are affected by several impairments that significantly limit the transmission rate. In fact, besides the intrinsic (technological) bandwidth limitation, due to transmitter capabilities, underwater acoustic channels are also characterized by severe multipath delays and highly selective frequency responses, as well as non-negligible propagation delays due to the relatively slow speed of sound in water [7]. The UWAC propagation scenario exhibits a peculiar time-varying behavior [8,9], due to sea temperature and salinity gradients, sea stream, and wind speed, as well as relative positional changes due to motion of the terminals [10]. Multipath time delay and frequency-selective deep fading affect the UWAC channel in different ways, depending on the propagation environment, typically classified as shallow, medium or deep water [11]. The shallow water channel is characterized by several paths of comparable length, so the corresponding channel impulse response (CIR) presents a large number of close coefficients

with similar amplitudes. On the other hand, deep water CIR exhibits longer delays and small coefficients because of the long distances traveled by the signal echoes with respect to the line-of-sight wave. The medium water environment presents delay spread and path gain features between the above two cases.

The mitigation of channel impairments is fundamental to provide a reliable communication link. Typical countermeasures consider the use of equalization [12,13] in order to restore the quality of the received signal. Alternatively, adaptive modulation and coding schemes are developed in order to match the transmission parameters with the channel behavior [14–16].

In this regard, Orthogonal Frequency Division Modulation (OFDM) offers desirable features, like robustness to multipath delay spread, sufficiently high data rate, large bandwidth efficiency, and adaptivity of the modulation format to channel conditions [17]. In OFDM, modulated symbols are transmitted over different sub-carriers that, ideally, do not interfere with each other when propagating over frequency-selective and time-invariant channels, so that simple symbol-by-symbol detection can be adopted. In real cases, the potential inter-carrier interference can be mitigated through equalization [18]. Furthermore, it is worth highlighting that, in radio-frequency (RF) OFDM systems, the intersymbol interference (ISI) arising from multipath propagation is typically mitigated through the use of a cyclic prefix (CP), the length of which is set according to the channel delay spread [19]. On the other hand, the sudden variability of channel behavior, as well as the long CIR, make the use of a fixed CP impracticable in UWACs, since ISI may not be properly mitigated [20]. CP length optimization has also been investigated, but mainly in reference to RF OFDM scenarios [21,22]. However, such solutions cannot be employed directly in UWACs, due to the significant differences between RF and underwater acoustic channels. Therefore, channel issues must be addressed in a different fashion. In this regard, OFDM adaptivity to the medium conditions is a relevant feature for the harsh propagation scenario offered by UWACs, allowing a significant increase in transmission rate. In [23], the authors propose an OFDM sub-carrier power optimization mechanism based on channel knowledge, achieved with a length-adaptive estimation technique. In [24], second-order statistics of the channel are exploited to derive the signal-to-interference-plus-noise ratio in each sub-carrier, so to realize adaptive coding and bit-power loading. An OFDM acoustic modem is presented in [25], with adaptive modulation being performed based on the channel delay and Doppler spread, measured by using chirp pilots. An information-dependent sub-carrier mapping is proposed for underwater video transmission in [26], so that important data are conveyed on the most reliable OFDM sub-channels, while less useful data are transmitted through the lower quality sub-carriers. The authors in [27] proposed a continuous phase modulation-based OFDM, outperforming standard implementations, and, thus, demonstrating itself to be more suited to UWACs. In [28], a novel fractional fast Fourier transform (FFT) OFDM system is presented, with amplitude shift keying (ASK) employed for sub-carriers' modulation. Specifically, the use of ASK achieves a better bandwidth efficiency with respect to other conventionally considered schemes and, furthermore, the use of fractional FFT in place of the conventional one allows problems related to carrier frequency offset to be more efficiently mitigated.

All the solutions for OFDM optimization rely on channel state information (CSI), made available from estimation or prediction. Overall, channel estimation typically considers a bi-directional transmission of pilot signals between the communication nodes, in order to measure the corresponding CIR [29] and feed back the channel coefficients. For example, in [30] a novel adaptive denoising scheme is proposed to achieve a reliable channel estimation in OFDM-based UWACs affected by strong noise. The authors in [31] propose an optimized pilot-assisted channel estimation for OFDM, even though performance was measured under the unrealistic assumption of ideal synchronization and Doppler compensation. Unfortunately, due to the low speed of sound, estimation procedures are time consuming and they must be performed frequently in order to keep the CSI updated. So, the communication rate may be significantly reduced. Furthermore, the available chan-

nel coefficients may be outdated, since, during the estimation process, the channel may have already changed [32]. In order to overcome such problems, channel prediction can be exploited in place of estimation. Predictors are typically based on channel statistical models [33] and are developed in different approaches, such as recursive least squares (RLS) [34] and deep neural networks [35]. Channel prediction does not impact on data rate since it allows the reduction of overhead information to be transmitted. On the other hand, if the predicted channel deviates from the real one, inaccurate system optimization occurs and problems of reliability may occur. The work in [36] considers, instead, the joint use of channel estimation and prediction. However, it is proposed for a single-carrier binary phase shift keying (BPSK) scheme where transmission parameter adaptation is not addressed. The worthiness of estimation and/or prediction is discussed in [37] by comparing minimum mean square error channel estimation and auto-regressive (AR) channel prediction. Furthermore, the authors propose a mechanism for bandwidth adaptation, in order to provide channel-tailored performance and avoid interchannel interference (ICI). Finally, a possible approach for channel estimation may rely on deep-learning, as presented in [38]. The proposed solution is designed for Multiple-Input Multiple-Output (MIMO) OFDM systems, even though not specifically related to the underwater acoustic case.

*Motivation and Goals of the Work*

As previously stated, both OFDM and single carrier modulations require channel knowledge for transmitter processing (e.g., bit-loading, pre-equalization) or receiver processing (e.g., equalization), aimed at counterbalancing the propagation impairments. Most of the proposed solutions assume perfect channel knowledge at the receiver and/or transmitter side, so the achieved performance cannot match that expected in a real scenario. Actually, the accuracy of CSI unavoidably impacts on the communication performance in terms of trade off between reliability and data rate. However, to the best of our knowledge, only a few works in the literature address such an issue. Furthermore, despite the availability of several channel estimation and prediction techniques, we highlight the absence of a real protocol that drives the joint use of these approaches in an adaptive fashion, so as to maximize the performance in terms of bit error rate (BER), while minimizing the rate reduction caused by the overhead signaling. Finally, it is worth noting that CSI acquisition and transmission optimization are typically performed under the *optimistic* assumption of absence of interference. The presence of superposing external acoustic sources (mammals' communication, vessels' engines and so forth) may not be negligible when dealing with channel estimation and signal detection [39], and, hence, it should be conveniently taken into account.

Aimed at the above considerations, in this work we propose an adaptive OFDM scheme able to manage the cyclic prefix length adaptation by resorting to channel knowledge acquired thanks to both estimation and prediction techniques, with the goal of reducing ISI. The employment of estimation, followed by a channel predictive step, is peculiarly tailored to the underwater environment, since it allows reduction in overhead signaling due to training/pilot transmission. Such a saving is not negligible in an UWACs, characterized by very scarce resources. Summarizing, the main contributions of this work are the following:

- an adaptive mechanism to tune the OFDM cyclic prefix, based on the channel estimation phase;
- a channel tracking mechanism, based on channel estimation and a prediction stage, by resorting to Kalman filtering;
- a *two-side contemporary processing*, that is, transmitter and receiver signal processing is operated independently, thus avoiding the need for a feedback link to continuously communicate measures operated at the receiver side;
- a mechanism to decide when re-estimation is needed, based on channel behavior;
- a frame structure/protocol supporting the adaptivity of the whole system and allowing a practical implementation of both the channel estimation and prediction.

The approach herein presented differs from the literature as it effectively considers practical situations related both to interference and channel. In fact, these two impairments are generally considered but their temporal features are assumed as static or quasi-static. The proposed study investigates how to handle interference variations and channel variability, in terms of delay spread and amplitude coefficients. The resulting link optimization can also be seen as an implementation guideline for UWACs.

About the manuscript's organization, in Section 2 we introduce the system model, including the initial interference statistics acquisition, channel statistics estimation procedures, and the CP adaptation mechanism. Section 3 describes the core of the proposed communication protocol, including the two-side contemporary processing based on Kalman filtering prediction and data transmission phase. In Section 4 numerical results and performance comparisons are presented and discussed. Finally, Section 5 concludes the paper.

## 2. System Model and Connection Setup

Let us refer to an underwater link between two nodes, namely $u_1$ and $u_2$, where data communication can be potentially bidirectional. Therefore, $u_1$ and $u_2$ can act as both transmitter and receiver. The transmission is considered to be OFDM-based and frame oriented, with frame length defined as $T_f$. By definition, each frame is then organized in slots of duration equal to $T_s$. Due to the time-varying nature of the underwater acoustic channel, we assume $T_f$ to be shorter than the channel coherence time $T_{coh}$, so that $T_f \leq T_{coh}$.

The interaction between $u_1$ and $u_2$ is organized in two different phases, referred to as connection setup (CS) and established connection (EC), respectively, described in Figure 1. During CS, both the nodes are involved in interference and channel acquisition, including the evaluation of the statistical properties of both interference and channel, necessary for the initial communication setup and for managing the whole communication. The EC stage instead concerns data transmission and detection, with some (small) time intervals dedicated to refreshing the channel and interference statistics. The stages of CS and EC are detailed in the following:

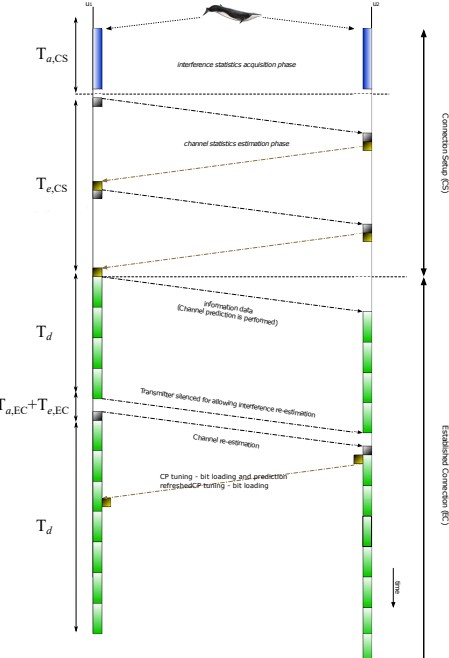

**Figure 1.** Graphical description of the three phases for interference and channel estimation and data transmission/detection during CS and ES.

Before starting the description of the whole signaling, we report in Table 1 all the time interval definitions we use in this work.

**Table 1.** List of variables related to time intervals.

| Symbol | Definition |
|---|---|
| $T_f$ | Transmission frame length |
| $T_s$ | Slot time of a frame, equating the OFDM symbol length (including CP) |
| $T_c$ | Sampling time |
| $T_x$ | OFDM symbol length (without CP) |
| $T_{CP}$ | OFDM CP length |
| $T_{a,CS}$ | Interference acquisition time during CS |
| $T_{a,EC}$ | Interference acquisition time during EC |
| $T_{e,CS}$ | Channel acquisition time during CS |
| $T_{e,EC}$ | Channel acquisition time during EC |
| $T_d$ | Frame portion of time dedicated to data transmission during EC |

*2.1. Interference Statistics Acquisition Stage*

As previously stated, and without loss of generality, let us refer to $u_1$ and $u_2$ as the transmitting and receiving nodes, respectively, placed at the left and right side of the time diagram in Figure 1 (however, transmitting and receiving roles can be interchanged). The first stage of CS is the acquisition of interference (represented by the vertical blocks in Figure 1). This stage must be used by both the nodes interested in setting up the communication, and it deals with the acquisition of interference statistics that can require a long time, and even multiple frames. It is worth noting that, depending on the (possibly) different propagation conditions, the result of interference acquisition is different at the transmit and receive sides. During interference statistics acquisition, lasting $T_{a,CS}$ seconds, no transmission is acted on by the nodes that remain in listening mode. Hence, the continuous-time received signal is:

$$r_{a,\text{CS}}^{(u_i)}(t) = z^{(u_i)}(t) = w^{(u_i)}(t) + \chi^{(u_i)}(t), \quad i = 1, 2 \tag{1}$$

where $w^{(u_i)}(t)$ is the zero mean $\mathcal{N}_0$-variance Additive White Gaussian noise at node $u_i$ and $\chi^{(u_i)}(t)$ is the possibly present interference due to acoustic sources (e.g., mammals and/or ship engines) plus the ambient noise described in [40]. Moreover, the term $\chi^{(u_i)}(t)$ in Equation (1) can be detailed as follows:

$$\chi^{(u_i)}(t) = \sum_{n=1}^{N_I} \psi_n(t) * h_{n,u_i}(t) \tag{2}$$

with $\psi_n(t)$ being the *n*-th interfering source (in a number of $N_I$), $*$ the convolution operator and $h_{n,u_i}(t)$ the impulse response describing the channel from the *n*-th interference source to the node $u_i$. It is important to underline that both $\chi_n(t)$ and $h_{n,u_i}(t)$ change, frame by frame, due to sea stream changes, the Doppler effect and multipath changes. This is the reason for performing long acquisition during the connection setup.

Furthermore, it is important to emphasize that, at regime, the time-varying nature of interference may be considered limited within multiple frames, thus meaning that statistical features do not change among some consecutive frames, let us say in a number of $Q_a$. At first sight, such an assumption may appear unreasonable. In fact, while for periodic and quasi-periodic interferences (e.g., engine of a ship) it is simple to prove that statistics do not change within a certain number of frames, a really different scenario is met when dealing, for instance, with mammal sounds. However, in this regard, the analysis reported in [41] proves that sound capture lasting in the order of hundreds of milliseconds allows a reliable collection of the statistical features of several interference sources, including also the sporadic ones (e.g., mammals).

So, during the first $T_{a,\text{CS}} = Q_a T_f$ seconds, both the nodes can proceed with the estimation of interference autocorrelation by simply correlating the $T_c$-sampled received signal as follows:

$$c_z^{(CS)}[m] = \frac{1}{2K_{a,\text{CS}} - 1} \sum_{p=1}^{K_{a,\text{CS}}} r_{a,\text{CS}}^*[p] r_{a,\text{CS}}[p+m] \qquad (3)$$

where the symbol $^*$ means conjugation and $K_{a,\text{CS}} = Q_a T_f / T_c$ is the length of autocorrelation. We remark that Equation (3) generically refers to the interference acquired by any of the considered nodes, but, of course, the measured values may not be the same due to different propagation conditions (that is, $h_{n,u_1}(t) \neq h_{n,u_2}(t)$, as detailed in Equation (2)). As the interference changes in time, statistics must be necessarily refreshed. Therefore, during EC, a time interval $T_{a,\text{EC}} = M_a T_s$ is reserved for autocorrelation update by using Equation (3), with $M_a$ being the number of frame slots reserved for such an operation. Given $K_{a,\text{EC}} = M_a T_s / T_c$ as the number of samples considered for the update, we have:

$$c_z^{(EC)}[m] = \frac{1}{2K_{a,\text{CS}} - 1} \sum_{p=1}^{K_{a,\text{CS}}} r_a^{(\mu)*}[p] r_a^{(\mu)}[p+m] \qquad (4)$$

being $r_a^{(\mu)}[p]$ defined in the vector format as:

$$\mathbf{r}_a^{(\mu)} = \left[ \mathbf{r}_a^{(\mu-1)}(K_a : K_{a,\text{CS}}) \; \mathbf{r}_a \right] \qquad (5)$$

where the superscript $(\mu)$ indicates the ordered number of updating procedure, the position $\mathbf{r}_a^{(0)} = \mathbf{r}_{a,\text{CS}}$ is assumed, and $\mathbf{r}_a^{(\mu-1)}(K_a : K_{a,\text{CS}})$ refers to the elements of the vector $\mathbf{r}_a^{(\mu)}$ ranging from the $K_a$-th till to the $K_{a,\text{CS}}$-th one.

Interference acquisition during CS may last several frames since sufficiently large statistics must be collected to reliably initialize the communication. On the other hand, interference acquisition at EC stage is performed to achieve a partial update of the statistics available from CS. So, its duration is only limited to some frame slots. A different way of defining Equation (5) is that the new acquisition of interference $\mathbf{r}_a$ (that is, the sampled version of the signal already described in Equation (1) during CS stage) gathers the most recent $K_a$ samples that update the vector $\mathbf{r}_a^{(\mu)}$, while the oldest $K_a$ ones are removed from the sequence $\mathbf{r}_a^{(\mu-1)}$.

### 2.2. Channel Model, Estimation and Statistics Evaluation

After interference acquisition, both the nodes must proceed with channel estimation as the second phase of CS, depicted in Figure 1. So, node $u_1$ starts sending some known pilot symbols to $u_2$, that replies with an identical transmission back to $u_1$. The pilot is modeled as:

$$x(t) = \sqrt{E_{s_0}} s(t) \qquad (6)$$

where $E_{s_0}$ is the associated energy and $s(t)$ is the equivalent baseband signal given by:

$$s(t) = \frac{1}{\sqrt{2\pi\alpha}} e^{\frac{-(t - \frac{T_s}{2})^2}{2\gamma}}, \quad 0 \leq t \leq T_s \qquad (7)$$

where $\gamma$ takes care of signal shape (and, consequently, its bandwidth), while $\alpha$ is set so to have a unit energy signal $s(t)$, since the signal $x(t)$ in Equation (6) has $E_{s_0}$ energy. The pilot length is supposed to be equal to $T_s$, with $s(t)$ being a single carrier signal, so as to allow the estimation of the whole channel. The choice of a Gaussian-like shape, described in Equation (7), is not mandatory and depends on the transmitting device capability to generate different shaping signals. Changing the signal shape (from Gaussian-like to Nyquist-like) would reflect on performance variations that are negligible if the bandwidth

and transmit power levels do not change. If the term $\gamma$ is sufficiently large, the signal is wide in bandwidth, since in the time domain it presents quick amplitude variations. At this point, such single-carrier pulse sent for estimation risks appearing a bit counter-intuitive since we base the communication on OFDM. However, by using such a pilot, we focus on a *whole* channel impulse response in order to optimize the cyclic prefix. Additionally, through the use of filter-bank (equating the number of OFDM sub-carriers) we are able to estimate the time response of each sub-channel. This scheme does not avoid the frequency domain description of the channel through the Discrete Fourier Transform (DFT).

With such bidirectional pilots transmission, each node can perform its own channel estimation. Especially regarding multipath, signal propagation along the two communication directions is different, that is, the channel is, in general, not reciprocal [42]. As a consequence, channel estimates available at the link sides may be different as well. However, since the largest part of channel energy is generally carried by the first path [43], reciprocity can be reasonably assumed. Moreover, what is important to underline is that, when reciprocity cannot be assumed, the channels are different. In other words, forward and backward channels can be different. However, it is highly probable that the differences do not impact on performance in a severe way. Hence, the conclusion is that reciprocity is worth it, especially when very particular propagation scenarios are not present [42], like, for example, obstacles that reflect a signal in a specific direction and block the signal in the opposite direction.

In order to describe the received signal during pilot transmission, let us now introduce the channel model. In detail, the underwater acoustic channel is typically affected by a frequency-selective fading phenomenon that scatters the energy of the transmitted signal over a (usually) not so small number of paths generated by reflections from ground and sea surfaces, or due to propagation effects that can be described as curved rays. By also taking into account its time-variability, the channel can be modeled according to the following expression [40]:

$$h(t;\tau) \triangleq \sum_{d=1}^{\rho(t)} \beta_d(t)a_d\delta(t-\tau_d(t))e^{j2\pi\nu_d t} \tag{8}$$

where $a_d$ is the complex coefficient accounting for losses over the $d$-th path [40], $\rho(t)$ is the time-varying number of paths, each one characterized by a propagation delay $\tau_d(t)$. Furthermore, $\beta_d(t)$ considers the shadowing effect of sea stream, dives, and fish schooling, according to [44], while $\nu_d = v\cos(\phi_d)f_c/v_c$ takes care of the Doppler effect through the direction $\phi_d$, with $v$ being the node speed, $v_c$ the speed of sound and $f_c$ the reference frequency.

For some time intervals and propagation scenarios, the channel can be considered time-invariant. As outlined at the beginning of this section, we assumed $T_f \leq T_{coh}$ to reasonably meet such conditions. Hence, we have $\rho_p(t) = \rho$ within a frame, and the received analog signal obtained from the transmission of a pilot can be represented as:

$$r_e(t) = \sum_{d=1}^{\rho} \beta_i d(t)a_d x(t-\tau_d)e^{j2\pi\nu_d t} + z(t) \tag{9}$$

that represents an exhaustive model, since it includes several propagation effects, from multipath to shadowing, due to obstacles or fish schooling.

### 2.3. Channel Statistics Estimation

During CS, channel statistics estimation may take a long time to achieve reliable information. Indeed, transmitting several symbols in a short time interval does not allow for observation of remarkable channel changes. On the other hand, using long guard intervals (in the order of the frame time $T_f$) between two consecutive pilots allows the receiving nodes to acquire more significant channel statistics. In principle, there are several ways to estimate the channel coefficients starting from the received signal in Equation (9).

Based on the a priori knowledge of the nodes about the signal shape $x(t)$, the channel autocorrelation can be retrieved from an observation time $T_{e,\text{CS}} = Q_e T_f$, with $Q_e$, being the number of frames spent for channel acquisition. In fact, similar to the interference statistics estimation in Equation (3), the channel autocorrelation can be evaluated as:

$$c_h^{(CS)}[m] = \frac{1}{E_{s0}(2K_{e,\text{CS}} - 1)} \sum_{p=1}^{K_{e,\text{CS}}} r_{e,\text{CS}}^*[p] r_{e,\text{CS}}[p+m] \tag{10}$$

with $K_{e,\text{CS}} = Q_e T_f / T_c$ being the number of samples used for estimation.

Analogously to the interference autocorrelation estimation, during EC, channel auto-correlation, updates can be performed exploiting a shorter time interval $T_{e,\text{EC}} = M_e T_s$ ($M_e$, being the number of employed pilots/frame slots) and, according to:

$$c_h^{(EC)}[m] = \frac{1}{E_{s0}(2K_{e,\text{CS}} - 1)} \sum_{p=1}^{K_{e,\text{CS}}} r_e^{(\mu)*}[p] r_e^{(\mu)}[p+m] \tag{11}$$

being $r_e^{(\mu)}[p]$ defined in vector format as:

$$\mathbf{r}_e^{(\mu)} = \left[ \mathbf{r}_e^{(\mu-1)}(K_e : K_{e,\text{CS}}) \; \mathbf{r}_e \right], \tag{12}$$

where $K_e = M_e T_s / T_c$ is the number of most recent autocorrelation samples. Similarly to the analysis reported in Equation (5), the superscript $(\mu)$ in Equation (12) indicates the number of updating procedure, the position $\mathbf{r}_e^{(0)} = \mathbf{r}_{e,\text{CS}}$ is assumed, and $\mathbf{r}_e^{(\mu-1)}(K_e : K_{e,\text{CS}})$ refers to the elements of the vector $\mathbf{r}_e^{(\mu)}$ ranging from the $K_e$-th till to the $K_{e,\text{CS}}$-th one. Finally, $\mathbf{r}_e$ defines the new estimation used for updating the channel autocorrelation, that is the sampled version of Equation (9).

### 2.4. Cyclic Prefix Length Adaptive Tuning

The cyclic prefix is a fundamental element in OFDM, allowing the system to be (hopefully) ISI- and ICI-free [45]. Due to possible relative motion between nodes, as well as sea stream changes, the channel results are unavoidably time-varying over consecutive frames. Therefore, the use of a fixed CP length during the whole communication may be ineffective to counterbalance multipath. On the other hand, the capability of tuning the CP length allows the transmission to be adapted to the propagation conditions.

In this regard, we have the number of sub-carriers $N_{SC}$ employed for OFDM signaling defining the symbol length $T_x = 1/\Delta_f$, where $\Delta f$ is the sub-channel width obtained by partitioning the whole system bandwidth $B$ in $N_{SC}$ portions, so that $\Delta_f = B/N_{SC}$. However, when CP is considered, the whole OFDM symbol time becomes $T_s = T_x + T_{CP}$, where $T_{CP}$ is the time length of cyclic prefix that must be set according to the channel delay spread $\tau_{ds}$. So, the OFDM symbol length corresponds to the slot time length $T_s$ within a communication frame. Here, we recall that the delay spread is a measure of the length of the channel response, that is, the time difference between the earliest channel path and the last one. Hence, analytically speaking, $T_{CP}$ value can be a multiple integer of delay spread. Of course, the use of CP unavoidably causes transmission rate reduction. In order to provide a theoretical example, if the system must be set up to grant a rate of $R$ bits/s, the information bits must be allocated so that $\sum_{c=1}^{N_{SC}} b_c = RT_s$, where $b_c$ is the number of bits allocated on the $c$-th sub-channel. The presence of CP entails $T_{CP} > 0$, leading $T_s$ to increase. From this fact, it appears evident that when the OFDM symbol length grows, due to delay spread (cyclic prefix), the use of richer modulation formats is required in the sub-channels to achieve the target data rate. As a consequence, higher power must be spent in order to guarantee robustness to ISI and ICI. So, communication reliability is paid for in terms of power efficiency.

The channel knowledge is the key to estimate the delay spread, that can be obtained by measuring the *memory* of the channel; that is, how long the effect of a signal emission is present at the receiver. Based on Equation (9), describing the pilot received during the estimation phase, it suffices to measure the cross-correlation between the emitted signal and the received one in order to test the signal time spread due to propagation. Analytically speaking, the cross-correlation is defined as:

$$y_{r_{es}}[m] = \frac{1}{E_{s0}(2T_s/T_c - 1)} \sum_{p=1}^{T_s/T_c} r_e^{(\mu)*}[p]s[p+m] \tag{13}$$

where $y_{r_{es}}[0]$ corresponds to the signal energy on the first path. Then, by means of a threshold $\vartheta$, $0 < \vartheta < 1$, we can also define a reference energy level expected for the signal on the last path, expressed as a fraction, $y_{r_{es}}[0]$. In other words, once $y_{r_{es}}[0]$ is selected as the cross-correlation value referred to the first path, we define, as delay spread, the time to achieve a percentage in terms of cross energy equal to $\vartheta y_{r_{es}}[0]$. Hence, formally we have:

$$\tilde{\tau}_{ds} = T_c \arg \min_{m=1,\dots,T_s/T_c} |y_{r_{es}}[m] - \vartheta y_{r_{es}}[0]| \tag{14}$$

returns the estimate of delay spread.

## 3. Established Connection Phase

As stated before, the channel is assumed to be static within a frame. During the EC stage, we can have two different types of frames. The first contains only data, thus, the time spent for data transmission, $T_d = M_d T_s$, with $M_d$ being the number of data symbols per frame, equates to the entire frame length; that is $T_f = T_d$ (note that, since data are transmitted only during EC, we neglect the use of the corresponding subscript on $T_d$ in order to simplify the notation). Such a frame can be easily recognized, highlighted in green in Figure 1. In this sense, the transmission of frames is continuous, while the channel prediction mechanism proceeds in the background. However, still from Figure 1, it is possible to appreciate that some frames are different, since they are organized in three different fields. The first one, of $T_{a,EC} = M_a T_s$-length, is dedicated to the interference acquisition. The second one, of $T_{e,EC} = M_e T_s$-length, is used for channel estimation purposes, while the last field, of $T_d$-length, is dedicated to sending information data. The whole time duration of a frame is $T_f = T_{a,EC} + T_{e,EC} + T_d$. Hence, for a fixed $T_f$, spending time on interference and channel statistics update unavoidably leads the data transmission time to be reduced within a frame.

### 3.1. Channel Re-Estimation and Prediction

The channel estimation can be performed in the discrete frequency domain. Referring to the model presented in Equation (9), by means of the DFT, the received pilot signal is:

$$R_k = H_k X_k + Z_k \quad 0 \le k \le N_{SC} - 1 \tag{15}$$

where $X_k$ and $Z_k$ are the DFT of the sampled version of $x(t)$ and $z(t)$, respectively, $H_k$ is the channel representation in the discrete frequency domain, and the number of samples to compute DFT is exactly the number of sub-channels $N_{SC}$. According to the orthogonal projection lemma criterion, detailed in [46], the Minimum Mean Square Error estimation of the channel can be obtained via the following relationship:

$$\tilde{H}_k = R_k \frac{X_k^* |H_k|^2}{E_{s0}|H_k|^2 + Z_k^2} \quad 1 \le k \le N_{DFT} \tag{16}$$

where, we recall, $E_{s0}$ is the energy associated to the training pilot, while $Z_k^2$ is the effect of disturbing signal $z(t)$. By observing Equation (16), we can argue that the $|H_k|^2$ values are

not available, since $H_k$ are the quantities to be estimated. This problem is solved by posing $|H_k|^2 = 1$, that leads to reliable estimates if $E_{s0} > Z_k^2$.

Starting from Equation (16), the channel evolution can be described by a V-order AR model [47–49] as follows:

$$\mathbf{h}_k(\ell) = \mathbf{\Psi}\mathbf{h}_k(\ell - 1) + \mathbf{z}(\ell) \tag{17}$$

where $\mathbf{h}_k(\ell)$ is a vector describing the channel coefficients related to the $k$-th sub-carrier, and $\mathbf{\Psi}$ is the $[V \times V]$ matrix related to the V-order AR model [47–49]. Based on such representation, we resort to Kalman filtering [50] to predict the channel evolution.

A possible drawback of this approach concerns the system complexity. In fact, tracking the channel evolution via Kalman filtering is not so costly for a single sub-channel, while it may become significant if a huge number of samples is considered. However, it is important to note that, at this stage, once the number of sub-carriers $N_{SC}$ is chosen, the Kalman-based prediction operates exclusively on $N_{SC}$ samples. So, the matrix dimensions used for handling the Kalman filtering is $[VN_{SC} \times VN_{SC}]$ and, even though it may appear to lead to a huge processing cost, it is composed of all $[V \times V]$ matrices on the diagonal, hence, presenting $V^2(N_{SC}^2 - N_{SC})$ zeros. Since no matrix inversions are present in the processing, the largest part of the computational cost is related to product and sums, to be performed with a processing time in the order of frame duration (hundreds of milliseconds).

Finally, we want to highlight here that channel prediction is operated both at the transmitter and receiver sides in place of channel estimation performed with overhead pilot signaling between nodes. Hence, prediction allows the reduction of latency, which represents a crucial issue in UWACs. Before proceeding, it is important to consider some key elements. First, it must be considered that the performance of the predictor (that is, the estimation accuracy) decreases in terms of prediction error variance when the coefficient to predict (in a time sense) is far from the last performed estimation. Hence, this fact suggests that, after predicting the channel for several frames, the receiver must proceed to re-estimate the channel by means of some pilot transmissions (as detailed in Figure 1). The motivation leading to requiring estimation after some prediction is twofold. First, after the CP tuning, it is expected that the delay spread may change. Second, the memory (in the sense of channel correlation) drops, thus meaning unreliability of the prediction. This is the reason for also introducing, in Figure 1, the pilot transmission during the EC stage, so as to provide the estimation to aid prediction. Therefore, a mechanism for re-estimating the channel and feeding the predictor with new estimated values is required.

In this regard, we need to regulate the switch from channel prediction to a new channel estimation. Let us consider $V_k$ as the length of the AR model, related to the $k$-th sub-channel. Please note that it is not assured that all the $V_k$ coincide. Moreover, delay spread may change during re-estimation, being smaller or larger with respect to the previously measured one. The delay spread being (considerably) changed indicates that the re-estimation was performed too late. On the other hand, if the changes measured are not sensible, it means that the re-estimation was performed early with respect to the changes. Hence, we introduce the variable $L_\epsilon$ to evaluate the measure of delay spread changes, initialized to a high value so that $L_\epsilon >> V_k$. Then, in order to switch from prediction to estimation, we consider re-estimation as performed after a number of frames $L_{adapt}$ given by:

$$L_{adapt} = \min\left\{\min_k\{V_k\}, L_\epsilon\right\} \tag{18}$$

thus, implicitly meaning that estimation is performed if $L_\epsilon$ frames have passed, or if the prediction becomes unreliable for at least one sub-channel ($\min_k\{V_k\}$). In other words, the event occurring first drives the need for a new estimation. It is important to note that, after a re-estimation is performed, if the channel delay spread has remained essentially the same, then $L_\epsilon$ is increased by a factor that we set as corresponding to 10% of its previous value. Otherwise, having delay spread as already changed suggests that we needed to re-estimate the channel earlier. Hence, $L_\epsilon$ is updated by decreasing it by 10%.

Finally, we would highlight that the proposed hybrid mechanism for channel estimation and prediction can be fruitfully exploited to deal with cross-talk mitigation, which represents a challenging issue for MIMO-OFDM systems. However, in such a more complex scenario, the sole channel equalization does not suffice to achieve good performance, since spatial ISI must also be mitigated. As a consequence, the number of channels to consider grows as well as the complexity of frequency and space equalization.

### 3.2. Information Data Stage and Detection

Having described the process related to interference and channel information acquisition, we now detail the data transmission stage, lasting $T_d$, as shown in Figure 1. Before symbol emission, bit-loading is performed. In this regard, we do not provide further details, since proposing a new mechanism was not our aim. However, it is fundamental to highlight that channel knowledge (and its reliability) is a key element to realizing bit-loading, especially when procedures based on *waterfilling-like* algorithms are considered [51]. Furthermore, channel knowledge is also necessary for signal detection. Specifically, we can express the OFDM data symbol as:

$$g(t) = \sqrt{E_{s_0}} \sum_{k=0}^{N_{SC}-1} G(k) e^{j2\pi kt\Delta f}, \quad 0 \le t \le T_x \tag{19}$$

where $T_x$ is the above-mentioned signal length without the insertion of CP. Moreover, the $G(k)$ term is related to the inverse-DFT of the symbol emitted on the $k$-th sub-channel. Note that symbols on different sub-channels may belong to different constellations (that is, the modulation order employed on sub-carriers may be different) if bit-loading is considered. The OFDM symbol is completed by CP insertion before transmission. If the CP length has been suitably adapted to completely avoid ISI, it follows that echoes related to the previously emitted symbol fall into the current symbol CP window, and, thus, do not affect the detection of carried data. Under such an assumption, the received analog signal related to an OFDM symbol after CP removal can be written as:

$$r(t) = \sum_{d=1}^{P} \beta_d(t) a_d g(t - \tau_d) e^{j2\pi \nu_d t} + z(t), \quad 0 \le t \le T_x \tag{20}$$

that collects the component $z(t)$ related to background noise and other external interference, and the signal echoes coming from the propagation over $P$ paths. Regarding this latter aspect, according to Equation (8), multipath was initially characterized by $\rho$ paths. However, since delay spread may be longer than the OFDM symbol duration $T_x$, it is likely that the late echoes fall within the next symbol CP window, while early arrivals are, instead, superposed to the currently received signal, thus giving rise to *auto-interference*. So, in Equation (20), we refer to $P \le \rho$ as the number of secondary paths acting as interference on the current symbol.

Finally, concerning detection, the receiver computes the DFT of the signal passed through analog-to-digital conversion. Then, by processing each sub-carrier component, the symbol $\hat{G}(k)$ (with $k = 0, 1, \ldots, N_{SC} - 1$) is decided according to the Maximum Likelihood criterion (by remembering that, on different sub-channels, we can have different modulation formats) [18].

### 3.3. Remark—Protocol Summary and Efficiency

The scheme here presented considers that, despite the CS stage, lasting $T_{a,\text{CS}} + T_{e,\text{CS}} = Q_a T_f + Q_e T_f$ seconds, being necessary, it is somewhat limiting from the point of view of transmission efficiency. Such a performance metric, referred as $\eta$, can be measured by considering the ratio for an assigned number of bits to be transmitted, between the time spent for an (ideal) OFDM communication with no connection setup, no estimation or prediction, and the presented case. Once set, the frame duration $T_f$, the number of slots composing a frame is equal to $M_f = T_f/T_s$. We recall from Section 2.4 that $T_s$ corresponds

also to the symbol duration, which, in turn, is dependent on the CP length $T_{CP}$. For the sake of simplicity, let us initially consider $T_{CP}$ as fixed (that is, the case where CP tuning is not performed), so that $T_s$ and $M_f$ result can be considered fixed as well. Hence, the time requested to transmit $N_b$ data bits in the ideal case, given $R$ as the transmission rate, is:

$$T_{N_b}^{(ideal)} = \frac{N_b}{R} \tag{21}$$

since no overhead information is considered during the communication. On the other hand, the use of the proposed protocol makes the data transmission time given:

$$T_{N_b}^{(proposed)} = (Q_a + Q_e)T_f + \frac{N_b}{R} + N_{occ}(M_a + M_e)T_f \tag{22}$$

thus, taking into account the $(Q_a + Q_e)$ frames used for CS and those related to EC, while $N_{occ}$ refers to the number of times interference and channel estimation refreshing is performed, allowing a reliable prediction to take place ($M_a$ and $M_e$ are recalled as the number of frame slots dedicated to interference and channel statistics update during the EC stage, respectively). The proposed mechanism efficiency is calculated from the ratio between the terms in Equations (21) and (22) that, after some mathematical manipulation, is expressed as:

$$\eta^{(proposed)} = \frac{1}{1 + \frac{R}{N_b}[(Q_a + Q_e)T_f + N_{occ}(M_a + M_e)T_f]} \tag{23}$$

where it is possible to observe that $\eta^{(proposed)}$ increases with the amount of data to be transmitted $N_b$. This is due to the fact that the growth of $N_b$ makes the CS stage duration ever shorter than the EC time.

Hence, the proposed method approaches the ideal case performance where no initial setup is considered. Furthermore, another aspect must be highlighted. By removing the initial assumption about fixed CP length, if $T_s$ changes due to the tuning of $T_{CP}$, the whole time to transmit data also changes. Specifically, independently of $N_b$, the growth of $T_{CP}$ leads $R$ to decrease and, interestingly, $\eta^{(proposed)}$ to increase. This is because a larger time is needed to transmit the information, and this growth is more important with respect to the time spent in CS. Another interesting result is the following. From Equation (22), the CS time is invariant, since $Q_a$, $Q_e$ and $T_f$ are fixed. So, CP adaptation impacts only on data transmission time, and this is true for both the ideal and proposed cases. Moreover, Equations (21) and (22) show the same dependency on $R$. The only difference regarding data transmission is given by the quantity $N_{occ}(M_a + M_e)$, that is only related to the proposed mechanism. Therefore, we can conclude from Equation (23) that the time spent in CS and re-estimating cannot be considered as marginal when $R$ is very high with respect to the data to be sent.

For the sake of comparison, we also discuss a reference case where channel estimation was systematically performed every frame, without resorting to prediction. Moreover, no CS and interference acquisition during EC were considered. Hence, at the beginning of each frame some pilots for channel estimation were sent and, due to the channel non-reciprocity, the transmitter had to wait for two-way propagation time $2T_{prop}$ before CSI was available. Note that $2T_{prop}$ is a function of the communication distance and may be very long. For example, in a 700 m link, the waiting time is about 1 s. In such a scenario, the data transmission time is calculated as:

$$T_{N_b}^{(estimation-only)} = \frac{N_b}{R} + \frac{N_b}{RT_f}(T_s + 2T_{prop}) \tag{24}$$

where the second term accounts for the time slot and propagation delay spent in transmitting the pilot symbols for channel estimation. So, based on Equations (21)–(24), we have that:

$$\eta^{(estimation-only)} = \frac{1}{1 + \frac{T_s + 2T_{prop}}{T_f}}$$

(25)

represents the spectral efficiency for the case of estimation performed at each frame. Specifically, it can be appreciated from Equation (25) that, in this case, spectral efficiency did not scale with $N_b$, thus, meaning that it was independent of the volume of data to be transmitted. Moreover, as expected, $\eta^{(estimation-only)}$ tended to zero when the propagation delay between the communicating nodes increased. So, we can conclude that the proposed mechanism, based on channel estimation and prediction results, is more convenient, in terms of efficiency, than a conventional approach, based only on the periodic channel estimation.

In order to summarize the whole mechanism, the flowchart reported in Figure 2 describes all the functional steps to be performed in the proposed communication framework. It is worth noting that, after a very large number of frames, referred as $Q_{max}$, where the communication is ongoing, it is reasonable to expect the channel to change significantly. In this case, it would be preferable to perform a complete refresh of interference and channel statistics by re-initializing the transmission with a new CS stage. Otherwise, as shown in Figure 2, a shorter acquisition and estimation during EC is sufficient to drive the Kalman filtering-based channel prediction.

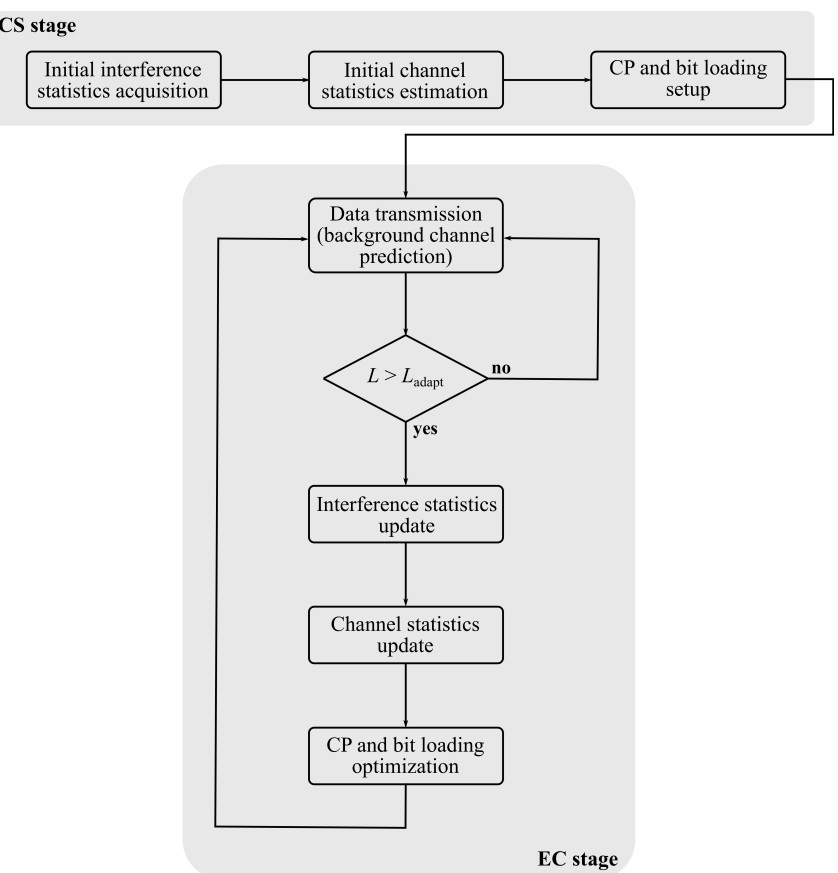

**Figure 2.** Protocol summary for CS and EC scenarios.

## 4. Numerical Results

In this section, we present the analysis of performance related to the proposed channel prediction-based adaptive transmission scheme. Simulations were performed by merging typical parameters of UWAC systems concerning the transmitter in terms of power and

bandwidth, and real data coming from measurement campaigns involving both channel and interference. Specifically, real multipath channel impulse responses, taken from the Watermark database [52], were considered to model the time-varying propagation. Channels were measured in a 740 m shallow water stretch of Oslofjorden, with sounding operated in a frequency range from 10 kHz to 18 kHz. The acquisition was performed by the authors of the measurement campaigns [52] by acquiring signals in raw data during a continuous time acquisition. Hence, consecutive time-variant channel impulse responses were measured and, due to the relative movement of transmitter and receiver, it was possible to infer that the average relative speed was 4 km/h. Furthermore, the interference generated by acoustic sources were taken from some recordings available in the literature [53] and directly added to the received signal. The results were strictly dependent on the communication bandwidth, chosen to be equal to 8 kHz. The other simulation parameters are summarized in Table 2.

**Table 2.** Simulation parameters.

| | |
|---|---|
| Bandwidth ($B$) | 8 kHz |
| OFDM sub-carriers ($N_{sc}$) | 128 |
| Transmit power ($P_{tx}$) | 183 dB @ 8V |
| Noise variance ($\mathcal{N}_0$) | $1.2 \times 10^{-23}$ W/Hz |
| Frame duration ($T_f$) | 20 ms |
| Slot duration ($T_s$) | 125 µs |
| Interference acquisition frames ($Q_a$) | 10 |
| Channel acquisition frames ($Q_e$) | 10 |

One of the goals of the analysis was to prove the effectiveness of a hybrid channel prediction-estimation-based approach with respect to a conventional pure estimation mechanism. So, as further detailed, the performance comparison between two such cases is also provided.

In order to, firstly, evaluate the accuracy of channel estimation, we report, in Figure 3, the mean square error (MSE) regarding the channel tracking as a function of the length of the statistics acquisition phase during the CS.

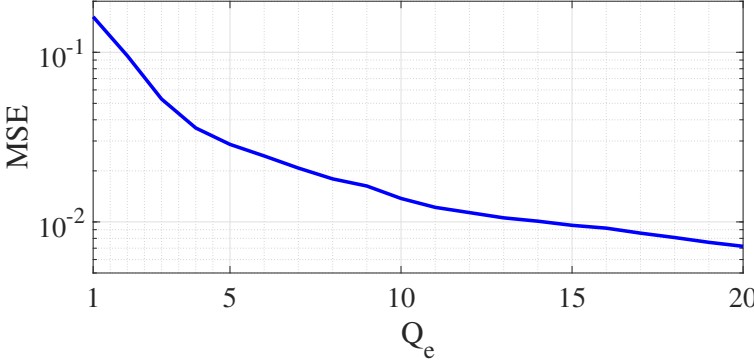

**Figure 3.** MSE for different lengths of acquisition phase time.

Overall, it can be observed that, the longer the initial acquisition was (measured in terms of number of frames $Q_e$), the lower the MSE was. In detail, the use of only a frame ($Q_e = 1$) did not lead a sufficiently reliable channel estimation and tracking with Kalman filtering, while, on the other hand, spending 20 frames for statistics acquisition allowed the MSE to be reduced by one order of magnitude.

Furthermore, regarding the tracking operation, we show in Figure 4 the value of $L_{adapt}$ in Equation (18) when different, consecutive channel realizations were considered. In particular, we discuss how the switching mechanism between tracking and estimation works. We recall that the need for re-estimation is driven by the minimum between the

memory of the auto-regressive model and the measurement of delay spread changes. In our simulation, we set the auto-regressive model memory as equal to 8 (frames) and, from Figure 4, it was possible to appreciate how such a value resulted, in general, as being the most appropriate one. In fact, most times, channel re-estimation results after 7 or 8 frames after prediction were exploited instead.

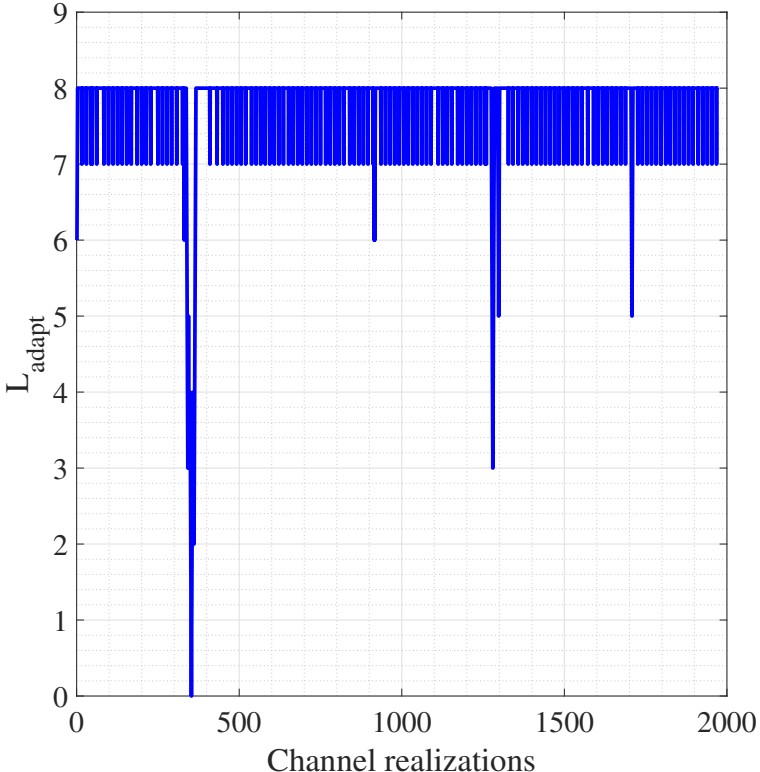

**Figure 4.** Values of $L_{adapt}$ in Equation (18) when different channel realizations are considered.

However, for some channel realizations exhibiting significant changes in propagation characteristics (e.g., delay spread), $L_{adapt} = 8$ was too high and the accuracy of prediction might lower. Hence, a more frequent re-estimation was required. The particular case where $L_{adapt} = 0$ refers to the occurrence where estimation is required on two consecutive frames.

Regarding performance comparison, we considered the competitor of the proposed adaptive transmission scheme to be one employing OFDM where pilot symbols are transmitted each frame to perform estimation, without considering prediction. Moreover, we assumed that, due to the channel non-reciprocity, estimation operated only at the receiver side, with channel coefficients being sent back to the transmitter via a bipolar modulation. Specifically, the information about the channel coefficients in the frequency domain was quantized with 32 bits before being fed back to the transmitter.

During this signaling phase, potential errors might occur, impacting on bit-loading process as well.

As a performance metric, we introduced $\sigma$ as the difference between the number of bits allocated via the ideal case (channel perfectly known at both the transmitter and the receiver) and the methods under investigation (that is, the proposed one and the frame-by-frame estimation with quantized feedback, respectively). The results are shown for two different channel realizations, referred as case (a) and (b) in Figure 5 , respectively. The numerical evaluation is reported on a per OFDM sub-carrier basis, that is, $\sigma_k = b_{ideal}(k) - b_{proposed}(k)$ are represented with red circled markers, while $\sigma_k = b_{ideal}(k) - b_{estimation-only}(k)$ are highlighted with blue markers ($k = 1, 2, \ldots, N_{sc}$). From Figure 5, it is possible to appreciate that the mismatch between ideal and proposed bit-loading was always equal to

zero in channel case (a), while only a few errors were made in case (b). Considering the scheme with frame-by-frame channel estimation, we can observe that there were several mis-allocated bits; in fact, the number of non-zero differences were 20 and 17 for cases (a) and (b), respectively.

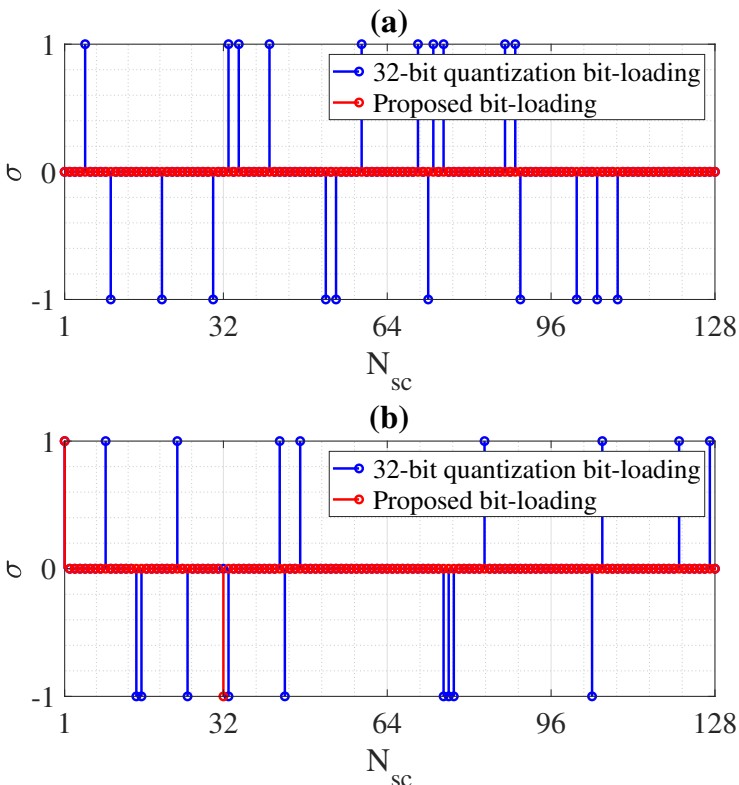

**Figure 5.** Difference between ideal bit-loading and the proposed scheme compared with the difference between ideal bit-loading and quantized information feedback link in channel realizations (**a,b**).

Mis-allocation in terms of bits leads to a situation that can be problematic. In fact, if the transmitter and receiver share mismatched channel information or, even worse, different information about the modulation format, the performance, in terms of rise in error rate, rapidly lowers. Such a result can be inferred from Figure 6, where we report the BER values by considering the evolution in terms of channel realization. In this regard, we want to emphasize that, in the bit-loading procedure we set an SNR-margin [54] granting a BER value of $10^{-6}$. By investigating Figure 6, for most of the channel realizations, target BER performances were granted by the proposed method. When this was not possible (due to time-varying propagation conditions not being perfectly compensated) the reliability lowered and BER increased to values around less than $10^{-4}$, with the exception of very few realizations where the channel was really bad. On the other hand, OFDM transmission with frame-by-frame estimation and quantized feedback was not able to achieve those values. In fact, the average BER was a bit higher than $2 \times 10^{-4}$, as is possible to appreciate by the distribution of blue markers in Figure 6.

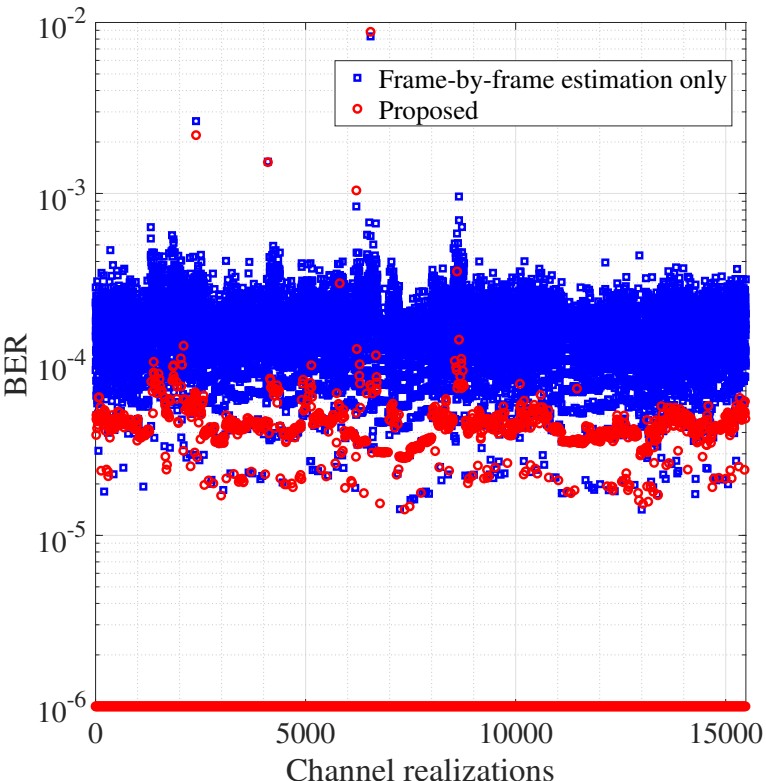

**Figure 6.** BER for different channel realizations for the proposed scheme and the case with frame-by-frame estimation and quantized feedback.

Another interesting aspect to evaluate is the impact of the number of FFT points $N_{SC}$, that is, the number of considered OFDM sub-channels, on the communication performance. In this regard, by referring to different values of $N_{SC}$, we report in Table 3 the average BER evaluated among all the channel realizations, as well as the maximum BER and minimum one.

**Table 3.** BER performance as a function of $N_{SC}$.

| FFT Points | Average BER | Maximum BER | Minimum BER |
|---|---|---|---|
| $N_{SC} = 16$ | $9.7 \times 10^{-6}$ | $5.6 \times 10^{-2}$ | $10^{-6}$ |
| $N_{SC} = 32$ | $6.1 \times 10^{-6}$ | $2.3 \times 10^{-2}$ | $10^{-6}$ |
| $N_{SC} = 64$ | $4.7 \times 10^{-6}$ | $1.4 \times 10^{-2}$ | $10^{-6}$ |
| $N_{SC} = 128$ | $3.6 \times 10^{-6}$ | $8.7 \times 10^{-3}$ | $10^{-6}$ |
| $N_{SC} = 256$ | $4.3 \times 10^{-6}$ | $1.1 \times 10^{-2}$ | $10^{-6}$ |

From the results, it can be emphasized that, using an increasing number of $N_{SC}$ led to a reduction in the average BER. This was true for $N_{SC} \leq 128$. However, when $N_{SC} = 256$, the average BER increased. We can explain this non-monotonic behavior as follows. When the number of sub-channels increases, the frequency domain description of the channel is more accurate, and the same is true for the prediction, since variations of the frequency response are measured with a dense sampling. On the other hand, having few FFT points (low values of $N_{SC}$) leads to a poor description of the channel in the frequency domain. So, channel changes in time are badly tracked and likewise for the equalization. The reason why for $N_{SC} = 256$ the average BER increased, is that the Doppler effect became more important, since the sub-channel bandwidth became narrower, and this induced mis-

detection pf events. By looking at the third column of Table 3, we note that the maximum BER followed the same behavior as the average BER, and such a high value was justified by the fact that sometimes the channel can change radically and very quickly. Finally, the fourth column of the table reports the same values for the minimum BER, since that was the target considered as the constraint for bit-loading.

Furthermore, another important result must be highlighted. We recall that the proposed transmission framework considers transmitting and receiving nodes as operating separately on interference acquisition and channel estimation, without any mutual feedback. So, it may happen that the bit-loading performed at the transmit side does not match with that expected at the receiver. This fact may lead signal detection to completely fail, being based on a symbol constellation that may be different from that actually employed for transmission.

However, the results in Figure 6 demonstrate the high reliability of the proposed hybrid channel estimation-prediction method, outperforming, also, the case where frequent channel estimation operated.

Such an advantage can be finally appreciated in terms of communication efficiency as well. By referring to $\eta$, as defined in Equations (23)–(25), for the transmission schemes under investigation, we show in Figure 7 that, when the message length increased, the OFDM system, where frame-by-frame estimation was performed, was unable to become more efficient, since the amount of pilot symbols to be sent increased with the number of transmitted frames. On the other hand, the proposed system paid off, in terms of efficiency, when the file was very short, since the time spent acquiring the statistics (CS stage) was long with respect to the transmission of a few bytes (10–30). However, the time spent in CS lost importance when the amount of information increased, as the largest part of the time was spent sending data (with an additional small percentage involved in re-estimating the channel).

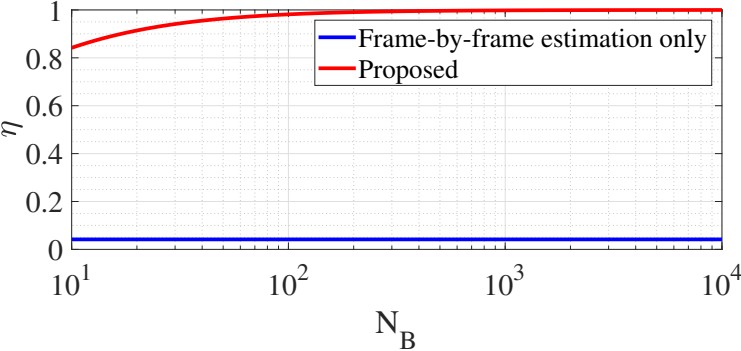

**Figure 7.** Efficiency for different lengths of data to be sent for the proposed scheme and the case with frame-by-frame estimation with quantized feedback.

## 5. Conclusions

In this contribution, we proposed an adaptive OFDM scheme, in which interference and channel statistics are initially acquired. Subsequently, in place of proceeding with very frequent channel estimation, as considered in most of the literature, we propose a mechanism to predict the channel and also a protocol to rule the re-estimation of the channel when necessary. This mechanism also solves the problem related to potentially outdated channel information due to long propagation delays. Simulation results demonstrated that the proposed approach is able to provide reliable channel tracking, also reflected in high performance in terms of BER and rate. Moreover, the designed communication protocol avoids the exchange of overhead information between transmitting and receiving nodes, thus, allowing the achievement of highly efficient communication, especially when the amount of data to be transmitted is large.



**Author Contributions:** Conceptualization, M.B.; Methodology, R.C.; Software, M.B.; Investigation, G.S.; Writing—original draft, A.P.; Writing—review & editing, A.P.; Supervision, M.B. All authors have read and agreed to the published version of the manuscript.

**Funding:** This work was partially supported by the European Union under the Italian National Recovery and Resilience Plan (NRRP) of NextGenerationEU, partnership on "Telecommunications of the Future" (PE0000001-program "RESTART").

**Conflicts of Interest:** The authors declare no conflict of interest.

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
