# Peer review of "On the Effect of Channel Knowledge in Underwater Acoustic Communications: Estimation, Prediction and Protocol"

_electronics, doi:10.3390/electronics12071552_

Round 1

Reviewer 1 Report

The authors present an adaptive OFDM scheme where channel state information is exploited to 11 optimize the cyclic prefix length, necessary to mitigate the multipath effect. A protocol is developed where channel estimation and/or prediction are jointly considered so as to provide a convenient trade-off between channel knowledge accuracy and data rate reduction. The simulations depict that estimating the channel through Kalman filtering is more efficient than feedbacking channel state information from the receiver to the transmitter.

The work is good with a detailed literature review and problem formulation. However, I have the following comments;

1) At the Rx the channel estimation and detection normally follow complex numerical evaluation, is it worth comparing the complexity comparison of the proposed scheme with the benchmark iterative procedure?

2) Is this filtering procedure applicable in the normal multiple antenna setup?

3) Equations 10 and 11 look exactly the same ;

4) The $r_e$ vector format in (12) is not clear; Is it the cancatenation of $r_e^u$f  elements ranging from $K_e and K_{e,C} $and $r_e$, 

5)  In equation (18) $V_k$ is a vector or kth value of $V$? what does $min_k{V_k}$ denote? Similar comment for $min_k{L_k}$ 

6) How do you compare the OFDM setup in [1] below and what is the impact of FFT length on BER in your considered system model?

Ullah, Arif, et al. "Soft-Output Deep-LAS Detection for Coded MIMO System: A Learning-Aided LLR Approximation." (2022).

Reviewer 2 Report

1. Grammar needs to be improved, there are many mistakes for example LINE 5 "basing" LINE 16 "more worth" LINE 30 "fitting for" LINE 96 "kept" to mention a few. Also, sentences must be short and concise. 

2. The abstract falls short of mentioning the novelty of work.

3. "Simulation Setup" section needs to be added. The experimental setup details are very vague. 

4. Doppler effects are ignored in the simulation setup section.

5. Conclusion needs to be rewritten. 

Round 2

Reviewer 2 Report

All my concerns have been addressed however the resubmitted manuscript is missing references and must be fixed in proofs.